# Exploring the Impact of Smartphone Addiction on Risk Decision-Making Behavior among College Students Based on fNIRS Technology

**DOI:** 10.3390/brainsci13091330

**Published:** 2023-09-15

**Authors:** Xiaolong Liu, Ruoyi Tian, Huafang Liu, Xue Bai, Yi Lei

**Affiliations:** 1Institute of Brain and Psychological Sciences, Sichuan Normal University, Chengdu 610066, China; ruoyitian98@163.com (R.T.); 13053664612@163.com (H.L.); baix0831@163.com (X.B.); 2The Clinical Hospital of Chengdu Brain Science Institute, MOE Key Laboratory for Neuroinformation, University of Electronic Science and Technology of China, Chengdu 611731, China

**Keywords:** smartphone addiction, risk decision-making, college students, fNIRS

## Abstract

Smartphone Addiction is a social issue caused by excessive smartphone use, affecting decision-making processes. Current research on the risky decision-making abilities of smartphone addicts is limited. This study used the functional Near-Infrared Spectroscopy (fNIRS) brain imaging technique and a Sequential Risk-Taking Task experimental paradigm to investigate the decision-making behavior and brain activity of smartphone addicts under varying risk levels. Using a mixed experimental design, the research assessed decision-making ability and brain activation levels as dependent variables across two groups (addiction and control), two risk amounts (high and low), and two outcomes (gain and loss). The study included 42 participants, with 25 in the addiction group and 17 in the control group. Results indicated that risk level significantly impacted the decision-making ability of smartphone addicts, with high-risk levels leading to weaker decision-making ability and increased risk-taking. However, at low-risk levels, decision-making abilities between addicts and healthy individuals showed no significant difference. Furthermore, brain imaging results using fNIRS revealed stronger brain activation in the dorsolateral Prefrontal Cortex (dlPFC) region for smartphone addicts under loss outcome conditions, with no significant differences between the two groups in terms of brain activation at varying risk volumes. These findings are critical in promoting healthy smartphone use, guiding clinical treatment, and advancing brain mechanism research.

## 1. Introduction

As technology advanced and society progressed, smartphones have evolved from luxury items to daily necessities. They are now indispensable tools for communication, entertainment, and relaxation. However, the convenience smartphones bring to our lives is a double-edged sword, giving rise to the potential problem of smartphone addiction. Kwon et al. [1,2] characterized Smartphone Addiction (SA) as a social dysfunction stemming from excessive smartphone use, marked by poor tolerance, intense focus while using the phone, unease when the phone is not within reach, or the battery is dead, neglect of other activities, a subjective loss of control, and persistence in using the phone despite clear evidence of its harmful effects.

However, the definition of this problematic behavior is not universally agreed upon. Some researchers refer to it as Problematic Use of Smartphones, denoting inappropriate or excessive smartphone use that leads to negative consequences in personal and social life [3]. Others define it behaviorally as Compulsive Smartphone Use (CSU), describing individuals who carry their smartphones everywhere, compulsively and frequently checking them in various contexts [4]. In essence, despite variations in terminology, the core issue remains consistent across definitions—the physical and psychological harm resulting from smartphone overuse.

According to the 50th Statistical Report on the Development of China’s Internet, as of June 2022, mobile Internet users in China topped 1.051 billion, with a staggering 99.6% accessing the Internet via smartphones. These data underscore the prevalence of smartphones as the primary Internet access device for most people. An increasing number of individuals are struggling to regulate their smartphone usage, leading to overuse, dependence, and, in some cases, addiction [5]. This escalating trend of Smartphone Addiction (SA) is mirrored by a rise in associated psychosocial issues such as depression [6], sleep disorders [7], social anxiety disorders [8,9], and compromised decision-making capabilities [10], among other concerns. 

Decision-making, a high-level cognitive activity, plays a crucial role in psychosocial processes and shapes our course of action. In this process, individuals choose among competing behaviors based on the anticipated value or utility of the outcome [11,12]. As some researchers suggest, decision-making is an optimization process where individuals weigh the magnitude of gains and losses, the likelihood of outcomes, and their subjective expectations [13]. In essence, decision-making involves an individual evaluating multiple options and selecting the one that yields the most benefit.

Numerous factors can influence decision-making, including personality traits [14], gender [15], emotions [16], and risk [17]. Research indicates that as the level of risk escalates, individuals are less likely to act [17]. The degree of risk can sway decision-making, with individuals more likely to make risky decisions at lower risk levels and more conservative decisions at higher risk levels [18].

As a complex cognitive process, decision-making requires the involvement of several brain regions, with the Prefrontal Cortex (PFC) playing a pivotal role [19]. Studies have shown that different decision contexts elicit varying activation in the medial prefrontal cortex (mPFC) in the left and right hemispheres. For instance, the right prefrontal cortex shows more activation in response to favorable choices in ambiguous decision contexts, while its activation response to both choices is similar in risky contexts [20]. Furthermore, the dorsolateral prefrontal cortex (dlPFC) exhibits stronger activation responses to high-risk unfavorable choices compared to low-risk favorable choices in individuals with gambling addiction [19]. This underscores the prefrontal cortex as a key brain region in the exploration of decision-making functions. A deeper understanding of its role is vital to comprehend the decision-making process and the mechanisms associated with related psychological disorders. Therefore, it is crucial to delve deeper into the functionalities of the prefrontal areas in the context of decision-making.

Research has established that Smartphone Addiction (SA) correlates with impaired decision-making capabilities. Behavioral analysis conducted by Khoury et al. [10] demonstrated differences in decision-making abilities between smartphone addicts and healthy individuals across various contexts. In ambiguous decision-making situations, the addicted group displayed significantly weaker decision-making skills compared to their healthy counterparts. However, in risky decision-making contexts, no significant difference was observed between the two groups. In contrast, other studies have suggested that smartphone addicts exhibit inferior decision-making skills in risky situations [21]. Therefore, the question of whether the decision-making abilities of smartphone addicts differ from those of healthy individuals in risky situations remains unresolved in previous studies.

Earlier research has asserted that the degree of risk can influence individual decision-making [18]. However, the decision-making behavior of smartphone addicts under these conditions remains unclear. From a physiological perspective, heavier smartphone users have shown stronger connections from the ventral striatum (vSTR) to the ventral medial prefrontal cortex and weaker connections from the vSTR to the dlPFC [22]. An additional study on electrical skin responses revealed that smartphone addicts exhibited lower responses before making a disadvantageous choice and after experiencing a loss outcome but higher responses after receiving a reward outcome [10].

While there is growing interest in Smartphone Addiction (SA) and its impact on individual decision-making, existing research remains somewhat narrow and homogenous. Historically, investigations into the effects of SA on decision-making abilities have primarily focused on behavioral aspects, with few studies examining multiple perspectives or physiological factors. As previously mentioned, variations in risk levels can yield different decision-making behaviors, and these have not been thoroughly explored within the SA group. Furthermore, the examination of brain activity during decision-making in smartphone addicts requires enhancement. The aim of this study is to investigate whether smartphone addicts display different decision-making behaviors and brain neural activity at varying risk levels within a risky decision-making context. The study proposes two hypotheses: 

**Hypothesis 1.** 
*At high-risk levels, smartphone addicts demonstrate poorer decision-making abilities compared to their healthy counterparts. However, at low-risk levels, the decision-making abilities of the two groups are not significantly different. Here, low risk refers to the presence of one gold coin in a chest, and high risk refers to three gold coins in a chest. The disparity in decision-making ability is reflected in the total number of coins accumulated during the task.*


**Hypothesis 2.** 
*Notable differences are observed in the neural activity of the brain between smartphone addicts and healthy individuals at both high and low-risk levels. Smartphone addicts exhibit a heightened brain activation response to successful outcomes compared to unsuccessful ones. Here, successful outcomes refer to ceasing to open the box, while unsuccessful outcomes correspond to encountering demons.*


## 2. Materials and Methods

### 2.1. Participants

The number of participants in this study was determined through a Prior Power Analysis (PPA). Before the experiment, the required sample size was estimated to ensure sufficient statistical power. This estimate was based on the group design of similar experiments from previous studies, the behavioral effect size (partial η^2^ = 0.08) [23], and a criterion of statistical power not less than 0.8 [24]. It was determined that a total sample size of 18 (i.e., *n* = 9 per group) would allow the effect of the decision to be detected at a significance level of 0.05.

To satisfy this sample adequacy, a total of 42 participants were recruited from a university campus in Sichuan. This group comprised 14 males (average age 21 ± 1.89 years) and 28 females (average age 19.5 ± 1.59 years). All participants were right-handed, with normal or corrected vision, normal hearing, and no history of traumatic brain injury, addiction, neurological, or psychiatric disease. Recruitment was primarily conducted through online recruitment posters and on-campus advertisements. All participants voluntarily took part in the experiment, signed an informed consent form, and received payment upon completion. The experiments underwent ethical review prior to commencement.

Based on previous research on smartphone addiction (SA), this study employed the abbreviated version of the SA Scale to identify participants for both the SA group and the control group. Through random sampling, participants were recruited on campus and then categorized using a questionnaire. After the screening and classification process, the SA group comprised 25 participants (10 males and 15 females), while the healthy control group consisted of 17 participants (4 males and 13 females).

### 2.2. Experimental Design

This study employed a combination of behavioral measures and functional near-infrared spectroscopy (fNIRS) to investigate the decision-making performance and brain activation levels of smartphone addicts under different risk levels and outcomes using an open-box continuous risk decision-making task paradigm. A mixed experimental design of 2 (group: SA group, control group) × 2 (amount of risk: high risk, low risk) × 2 (outcome: gain, loss) was utilized, with group as a between-participants variable and the amount of risk and outcome as within-participants variables.

The dependent variables of the study included the probability of encountering a ghost (i.e., the proportion of ghost encounter trials to the total number of trials), the total number of gold coins obtained during the task, and the participants’ brain activation levels as measured by fNIRS.

### 2.3. Experimental Tools

#### 2.3.1. Smartphone Addiction Scale–Short Version (SAS-SV) [2]

The study utilized the Smartphone Addiction Scale–Short Version (SAS-SV), which consists of 10 questions rated on a 6-point scale ranging from 1 (strongly disagree) to 6 (strongly agree). All the questions are positively scored, indicating that higher total scores indicate a higher level of smartphone addiction. To evaluate smartphone addiction among Chinese college students, Zhao Hao et al. translated the SAS-SV scale into Chinese. The experimental results confirmed the effectiveness of the SAS-SV scale in measuring smartphone addiction among Chinese college students [25].

#### 2.3.2. Smartphone Addiction Scale for College Students (SAS-C) [26]

To improve the validity of participant grouping, the SAS-C was developed as an extension of the SAS-SV. This scale consists of 22 items and includes 6 factors: withdrawal behavior, social appeasement, emergent behavior, negative affect, app update, and app use. Total scores are calculated, with higher scores indicating a more severe level of smartphone addiction. Studies have demonstrated that the scale is both reliable and valid and can effectively measure the smartphone addiction status of Chinese college students [26].

#### 2.3.3. Positive Affect and Negative Affect Scale (PANAS) [27]

The Positive and Negative Affect Schedule (PANAS) was employed to evaluate positive and negative affect in this study. The scale consists of two dimensions, with each dimension comprising nine items rated on a 5-point scale. The positive emotions dimension includes items such as feeling exhilarated, excited, energetic, enthusiastic, proud, grateful, elated, active, and happy. On the other hand, the negative emotions dimension includes items such as feeling irritated, sad, guilty, frightened, irritable, ashamed, nervous, trembling, and fearful. Higher scores on the positive emotions dimension indicate a greater experience of positive emotions in the past month, while the same applies to higher scores on the negative emotions dimension. Previous studies conducted in China have validated the scale and confirmed that its various items possess good discrimination, making it an effective and reliable tool for measuring emotional well-being [27].

#### 2.3.4. Barratt Impulsiveness Scale-11 (BIS-11) [28]

The Barratt Impulsiveness Scale-11 (BIS-11) was utilized in this study to assess individual impulsivity. It consists of three dimensions: attentional impulsivity, motor impulsivity, and non-planning impulsivity. The scale includes 26 items rated on a 4-point scale (never, occasionally, often, always), with 11 items being reverse scored. The BIS-11 provides both a total score and scores for each dimension, with higher scores indicating higher levels of impulsivity. Previous research conducted with a sample of 2397 Chinese college students has demonstrated that the BIS-11 has high reliability and validity [29].

#### 2.3.5. Beck Depression Inventory-II (BDI-II) [30]

The BDI-II is a widely utilized scale designed to assess the severity of depression in individuals. It comprises 21 items that are divided into 2 factors: somatization-emotional factor and cognitive factor. Each item is rated on a scale from 0 to 3, resulting in total scores ranging from 0 to 63. The scoring categories are as follows: 0–13 indicating no depression; 14–19 indicating mild depression; 20–28 indicating moderate depression; and 29–63 indicating severe depression. Within the Chinese context, this scale has demonstrated commendable reliability and validity [31].

#### 2.3.6. Beck Anxiety Inventory (BAI) [32]

The BAI is a self-report scale specifically developed to assess an individual’s level of anxiety. It consists of 21 items that are rated on a 4-point scale, ranging from “none” to “severe”. The sum of the scores of the 21 items is then rounded using the formula *Y* = INT(1.19*x*) to obtain a standard score. Higher scores on the BAI signify higher levels of anxiety. Moreover, previous studies have confirmed that this questionnaire exhibits high reliability and validity in the context of Chinese-based research [33,34].

### 2.4. Procedures

This study was conducted in two stages: the scale administration stage and the experimental stage. Before recruitment, every participant was required to complete the abbreviated version of the SAS. Participants were subsequently categorized into two groups based on their scores: the SA group and the control group. Participants scoring more than 31 (males) and 33 (females) were classified into the addiction group, whereas those scoring less were placed in the control group [2]. Upon entering the laboratory, participants voluntarily provided informed consent for the experiment and proceeded to complete a series of scales, including the Demographic Questionnaire, SAS-C, PANAS, BIS-11, BDI-II, and BAI. These scales were utilized to collect demographic data and assess various psychological aspects of the participants. On the day of the experiment, participants also completed the Smartphone Addiction Scale for College Students (SAS-C) to ensure the consistency of their SA status before and after the experiment. The experimental design software, E-Prime 3.0, was utilized to construct the experimental paradigm. The stimuli were subsequently exhibited on a computer monitor with a resolution of 1920 × 1080.

Once the participants completed the scale ensemble, the behavioral and fNIRS experiments commenced. Throughout the experiment, fNIRS cerebral blood oxygen data were collected from the participants’ orbitofrontal cortex (OFC) and dorsolateral prefrontal cortex (dlPFC). Prior to the experiment, participants were informed that the gains in the experiment would be translated into real experimental rewards.

The formal experiment was divided into two blocks: the high-risk level and the low-risk level, with each block consisting of 50 trials, culminating in a total of 100 trials. In the low-risk level trials, participants earned one gold coin per chest, while in the high-risk level trials, they were awarded three gold coins per chest. The experiment commenced with a rest period, during which a blank screen was displayed for 60 s, succeeded by a 500 ms fixation point. Subsequently, eight boxes appeared on the screen, ushering participants into the decision phase. In this phase, they had to choose whether to continue opening boxes or to stop. Opting to stop opening boxes signified preserving the gold coins acquired from previous openings. Conversely, continuing to open boxes presented the potential for additional gold coins but also the risk of encountering demons, which would result in the loss of all earnings in that round. Each box had a decision time frame of 0–2 s, and the decision phase for each trial lasted 0–16 s. This was followed by a 2–3 s blank screen and a result presentation screen. The result presentation screen displayed the outcomes for 3 s, succeeded by a 7 s blank screen before transitioning to the next trial. The entire experiment was conducted within a span of 30–40 min. The specific stages of the process are depicted in Figure 1.

### 2.5. fNIRS Data Acquisition

In this study, fNIRS data were acquired using a Wisetron portable near-infrared functional brain imaging device (NirSmart II). Participants were seated in front of a computer in the laboratory during the data acquisition. The light source detectors were arranged with 8 light sources (Source) and 7 receivers (Detector), spaced 3 cm apart, forming 22 channels (Channel) that covered the OFC and dlPFC brain regions explored in this study (refer to Figure 2 for the channel layout). The channel positions were determined with reference to the 10–20 international standard lead system, with the probe positioned in the middle of the lower edge of the acquisition head cap at Fpz. fNIRS measurements were recorded for the measurement of Oxyhemoglobin (HbO) and Deoxyhemoglobin (HbR) in each channel within the brain region. The experimental sampling rate was set at 11 Hz.

### 2.6. Data Analysis

For the questionnaire results, an independent samples *t*-test was conducted, using ‘group’ as a between-group variable to investigate differences between the two groups. As for the behavioral outcomes, a repeated measures ANOVA was performed, with IGT scores serving as the within-participants variable to explore differences in stages. An independent samples *t*-test was also conducted, using ‘group’ as the between-groups variable to determine if there were any differences between the two groups in different contexts. Regarding the fNIRS results, the Homer 3.0 toolkit, based on MATLAB, was employed for fNIRS data preprocessing. Given that Oxyhemoglobin (HbO) is more sensitive to neural activity than Deoxyhemoglobin (HbR), HbO was selected as the indicator of blood oxygen level change. The preprocessing steps included the conversion of light intensity information into optical density data; artifact correction by channel; artifact correction via Spline; band-pass filtering at 0.01–0.5 Hz to eliminate irrelevant physiological noise; and the calculation of optical density information based on the modified Beers-Lambert law to convert the concentration change values of HbO and HbR. A general linear regression, combined with the experimental design model (General Linear Model, GLM), was used to estimate the relevant beta values. Block averaging was then performed. A mixed-measures ANOVA was subsequently used to explore the main effects and interactions. If interactions were present, simple effects analysis was employed for post hoc tests. When multiple comparisons were involved, a Bonferroni correction was applied to adjust the significance level of the test. Finally, the channel activation results were visualized using the EasyTopo software 1.0 [35].

## 3. Results

### 3.1. Demographics and Results of Each Scale

The demographics and results of each scale were analyzed using independent samples *t*-tests for both groups, with the findings presented in Table 1. Significant differences were observed in SAS-SV scores, SAS-C scores, negative affect scores, attentional impulsivity, depression scores, and anxiety scores, with the addiction group scoring significantly higher than the control group. 

### 3.2. Risky Decision-Making Behavior

This experiment aimed to scrutinize the risky decision-making behavior of the participants. The behavioral indicators included the total number of gold coins collected during the task and the probability of encountering ghosts (the ratio of ghost encounter trials to the total number of trials). A 2 (amount of risk: high risk, low risk) × 2 (group: SA group, healthy control group) repeated measures ANOVA on the total number of gold coins revealed a significant main effect of the risk level. Participants garnered significantly fewer gold coins at the low-risk level compared to the high-risk level (F(1, 40) = 2155.85, *p* < 0.001, partial η2 = 0.982). The results are depicted in Figure 3a. Tests for between-participant effects revealed significant main effect margins for groups, with participants in the addiction group having significantly fewer total gold coins than the control group (F(1, 40) = 3.993, *p* = 0.053, partial η2 = 0.091). A marginally significant interaction was found between risk amount and group (F(1, 40) = 3.875, *p* = 0.056, partial η2 = 0.982). A simple effects analysis was conducted and revealed (Figure 3b) that the total number of gold coins was significantly lower in the addiction group than in the control group at the high-risk level (*p* < 0.05). At the low-risk level, the total number of gold coins in the addiction group was not significantly different from the control group (*p* = 0.371).

Additionally, a 2 (risk amount: high risk, low risk) × 2 (group: SA group, healthy control group) repeated measures ANOVA on the probability of encountering ghosts found a significant main effect margin for the risk amount. Participants had a significantly higher probability of encountering ghosts at the low-risk level than at the high-risk level (F(1, 40) = 3.991, *p* = 0.053, partial η2 = 0.091). The between-participants effect test showed that the main effect of the group was not significant (F(1, 40) = 2.100, *p* = 0.155, partial η2 = 0.05). No significant interactions were found between risk volume and group (F(1, 40) = 0.745, *p* = 0.393, partial η2 = 0.018).

### 3.3. fNIRS Results

The correlated brain activation measurements of the participants are depicted in Figure 4 and Figure 5.

This study employed a 22-channel, repeated-measures ANOVA of beta values derived from GLM estimation for different conditions in various groups. The experimental design was a 2 (group: SA group, healthy control group) × 2 (risk amount: high risk, low risk) × 2 (outcome: gain, loss) for HbO. The findings are as follows: In channel 3, a significant main effect was observed for the amount of risk, with higher risk yielding significantly higher values (F(1, 40) = 6.131, *p* < 0.05, partial η2 = 0.133). However, the main effect of the outcome was not significant (F(1, 40) = 2.329, *p* = 0.135). In channel 4, the amount of risk had a significant main effect, with high risk yielding significantly higher values (F(1, 40) = 5.283, *p* < 0.05, partial η2 = 0.117). The outcome also showed a significant main effect, with lower return outcomes (F(1, 40) = 15.454, *p* < 0.001, partial η2 = 0.279). In channel 5, the amount of risk had a significant main effect, with high risk yielding significantly higher values (F(1, 40) = 11.034, *p* < 0.01, partialη2 = 0.216). The main effect of the outcome was marginally significant and lower for the return outcome (F(1, 40) = 3.842, *p* = 0.057, partial η2 = 0.088). A significant interaction was found between the amount of risk and outcome (F(1, 40) = 5.723, *p* < 0.05, partial η2 = 0.125). A simple effects analysis revealed that loss outcomes were significantly higher at high levels of risk (*p* < 0.05, Cohen’s d = −2.46), while at low levels of risk, there was no significant difference between gain and loss outcomes (*p* = 0.325). These results are illustrated in Figure 6a and Figure 7a.

In Channel 6, the main effect of the outcome was significant, with return outcomes being significantly lower (F(1, 40) = 9.391, *p* < 0.01, partial η2 = 0.19). In Channel 8, there was a significant interaction between the outcome and group (F(1, 40) = 8.632, *p* < 0.01, partial η2 = 0.177). A simple effects analysis revealed that under the loss outcome, the addiction group scored significantly higher than the control group (*p* < 0.05, Cohen’s d = 1.21). However, under the gain outcome, no significant difference was found between the two groups (*p* = 0.625). These results are depicted in Figure 6b and Figure 7b. In Channel 9, the main effect of the outcome was significant, with return outcomes being significantly lower (F(1, 40) = 10.082, *p* < 0.01, partial η2 = 0.201). In Channel 10, a significant interaction was found between the outcome and group (F(1, 40) = 4.658, *p* < 0.05, partial η2 = 0.104). A simple effects analysis showed that the addiction group had significantly lower gain outcomes compared to loss outcomes (*p* < 0.05, Cohen’s d = −1.93). No significant difference was found between the two outcomes in the control group (*p* = 0.465). These results are presented in Figure 6c and Figure 7c. In Channel 12, there was a significant interaction between outcome and group (F(1, 40) = 5.898, *p* < 0.05, partial η2 = 0.129). A simple effects analysis indicated that the addiction group had significantly lower gain outcomes compared to loss outcomes (*p* < 0.05, Cohen’s d = −2.5). However, no significant difference was observed between the two outcomes in the control group (*p* = 0.370). These findings are illustrated in Figure 6d and Figure 7d. In Channel 15, a significant interaction was noted between outcome and group (F(1, 40) = 4.562, *p* < 0.05, partial η2 = 0.102). A simple effects analysis demonstrated that under the loss outcome, the addiction group scored significantly higher than the control group (*p* < 0.05, Cohen’s d = 1.12). Yet, under the gain outcome, there was no significant difference between the two groups (*p* = 0.482). These results are shown in Figure 6e and Figure 7e. In Channel 18, the main effect of the risk amount was significant, with higher risk resulting in significantly higher outcomes (F(1, 40) = 4.276, *p* < 0.05, partial η2 = 0.097). However, the main effect of the outcome was not significant (F(1, 40) = 0.001, *p* = 0.976). In Channel 20, the main effect of the risk amount was significant, with larger risk amounts yielding significantly higher outcomes (F(1, 40) = 4.986, *p* < 0.05, partial η2 = 0.111). The main effect of the outcome was not significant (F(1, 40) = 0.035, *p* = 0.853). In Channel 22, the main effect of the risk amount was significant, with high-risk amounts resulting in significantly higher outcomes (F(1, 40) = 4.608, *p* < 0.05, partial η2 = 0.103). However, the main effect of the outcome was not significant (F(1, 40) = 0.004, *p* = 0.950). In the remaining channels, no significant main effects or interactions were detected.

## 4. Discussion

Smartphones have significantly impacted people’s lives. However, as usage increases, the issue of smartphone addiction has become increasingly prominent. Within the field of behavioral addiction research, the decision-making function, a crucial aspect of behavioral addiction exploration, is an important focus in the area of Smartphone Addiction (SA). However, most existing studies lack depth and do not explore SA in detail. Furthermore, few studies have investigated it at a physiological level. In this work, we utilized the Open Box Continuous Risk Decision-Making Task, in conjunction with functional near-infrared spectroscopy (fNIRS), to thoroughly examine the decision-making behaviors and brain neural activity of smartphone addicts. This was conducted in various decision-making contexts and at different risk levels from both behavioral and physiological perspectives.

Following questionnaire analysis, the findings of this study revealed that smartphone-addicted college students exhibit stronger negative emotions, higher impulsivity, and more severe levels of depression and anxiety compared to their healthy counterparts. This aligns with existing research findings that suggest these heightened emotional states or traits may lead individuals to seek escape from reality, thereby increasing their dependence on smartphones [36,37,38]. The pleasure and sense of fulfillment derived from smartphone use may also make individuals more reluctant to disconnect from their devices, thereby escalating their level of addiction.

This study’s behavioral findings suggested that college students addicted to smartphones demonstrate inferior decision-making capabilities and a heightened inclination towards risk-taking compared to their non-addicted peers. Specifically, at high-risk levels, smartphone addicts exhibited poorer decision-making skills and a stronger propensity for risk-taking than their healthy counterparts. However, at low-risk levels, no significant difference in decision-making was observed, which is consistent with the study’s hypothesis. Previous research has indicated that individuals tend to adopt more conservative decision-making strategies when confronted with higher risks [18]. However, addicts display contrary behavior. In related studies on behavioral addiction, it was discovered that for gambling addicts, high-risk positive reinforcement scenarios amplified their gambling cravings and the probability of gambling, irrespective of the risk amounts [39]. This implies that elevated risk levels intensify addicts’ cravings for addictive behaviors. Thus, it can be inferred that higher risk levels may incite smartphone addicts to indulge in more impulsive and risky behaviors [20].

This finding implies that smartphone addicts may impulsively opt for short-term beneficial but long-term detrimental choices, even when the probability of the options is clear. This confirms the ‘myopia’ in decision-making observed in the addicted population. This trait may be linked to the heightened reward sensitivity of smartphone addicts, who are more responsive to rewards [40]. Consequently, they are more likely to be drawn to high rewards in high reward-high punishment scenarios, leading them to obsessively make choices that are unfavorable in the long run. This characteristic is a defining feature of the decision-making behavior of the smartphone-addicted group.

From a neurophysiological perspective, two studies probing the behavioral manifestations of decision-making in smartphone addiction (SA) pinpointed the dorsolateral prefrontal cortex (dlPFC) and orbitofrontal cortex (OFC) as the primary brain regions activated. The research indicates that the left dlPFC in individuals is more attuned to loss outcomes at high-risk levels. However, at low-risk levels, no significant variance in brain activation between the two outcomes was observed. Both the OFC and dlPFC regions demonstrated increased sensitivity to high risk. Activation patterns within the dlPFC revealed that smartphone-addicted college students were more reactive to loss outcomes. Conversely, there was no notable difference in brain activation between the two groups when confronted with gain outcomes, and healthy individuals exhibited no significant disparities in brain responses between the two outcomes.

Although direct studies of brain activity in decision-making within the SA group are scarce, a physiological study on skin conductance demonstrated that smartphone addicts exhibited lower skin conductance responses to both adverse choices and loss outcomes compared to healthy individuals [10]. The findings of this research suggested that while healthy individuals are more sensitive to adverse choices, smartphone addicts show no preference, and are more responsive to loss outcomes. This could be attributed to the stronger impulsive traits seen in addicts, traits that healthy individuals typically do not possess [41]. Consequently, healthy individuals can maintain rationality throughout the decision-making process and exhibit greater sensitivity to adverse choices to maximize benefits, a trait not observed in addicts. Prior research has shown that addicts are more driven to achieve immediate gains, which may make them less tolerant of losing immediate rewards and, thus, more sensitive to loss outcomes [42].

In conclusion, smartphone addicts appear more prone to making detrimental decisions in risky contexts, while in ambiguous situations, they require more cognitive resources to make decisions. This finding corroborates previous research linking decision-making dysregulation with addictive behaviors. In high-risk scenarios, smartphone addicts tend to make unfavorable decisions and exhibit heightened sensitivity to loss outcomes. This supports prior research associating addictive behaviors with risky decision-making. 

This study holds significant implications for future research. It identified notable differences in behavior and brain activity between the smartphone addiction group and the control group, underscoring the detrimental effects of excessive smartphone use on physical and mental health. This insight inspires us to implement measures to enhance the supervision and management of teenagers’ smartphone usage, thereby safeguarding their physical and mental health. Furthermore, with continued research in this field, it could become possible to detect potential smartphone addicts early by observing brain activation during decision-making behavior, thereby enabling timely intervention and treatment. We believe this holds substantial implications for the management of smartphone addiction.

This study, while insightful, has a few limitations. First, we did not differentiate between various types of smartphone addiction. For instance, some individuals primarily use their smartphones for social networking [43], while others frequently engage in online gaming [44] or shopping [45]. We have identified significant differences in decision-making behavior and brain activity between smartphone addicts and non-addicts. Consequently, it would be compelling to investigate whether these differences are influenced by the specific content of smartphone usage. 

Second, our research primarily focuses on the frontal lobe, detailing the functions of regions like the dlPFC and mPFC. However, due to the constraints in coverage areas, we did not include some other brain regions. For instance, previous studies have demonstrated that the ventromedial prefrontal cortex (vmPFC) and pre-supplementary motor area (pre-SMA) are intricately linked to planning and decision-making activities. The pre-SMA, in particular, plays a critical role in human decision-making [46]. To enhance our research findings, future studies could broaden the coverage of brain regions and delve deeper into the activities of other brain regions involved in decision-making. 

Lastly, due to the limitations of fNIRS technology, we were unable to determine whether certain deep brain regions are implicated in decision-making behavior. Therefore, future research could employ high spatial resolution neuroimaging devices, such as functional magnetic resonance imaging (fMRI), to expand upon our findings and provide a more comprehensive understanding of brain activity.

Overall, this study experimentally investigated the decision-making behaviors and brain activation levels of smartphone addicts, revealing a connection between addictive behaviors, decision dysregulation, and risk. These findings offer valuable insights into better understanding the neural mechanisms involved in addictive behaviors and provide potential avenues for their treatment. Although there are some limitations, we believe that future research can fill the gaps in these technical methods to provide a deeper understanding.

## Figures and Tables

**Figure 1 brainsci-13-01330-f001:**
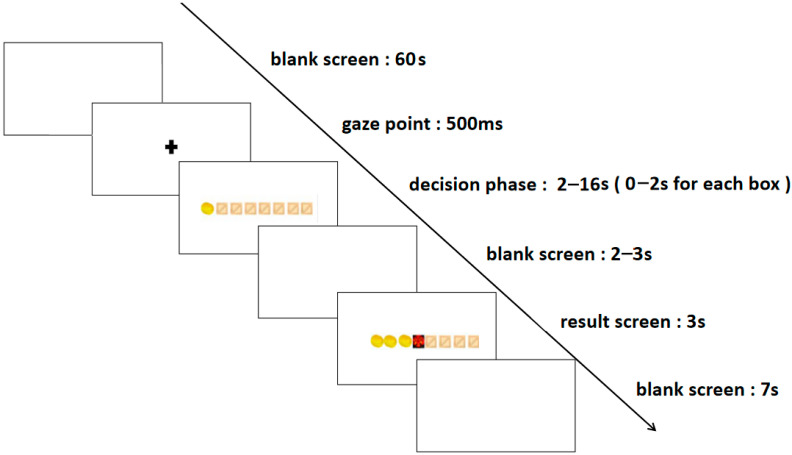
The experimental task flowchart.

**Figure 2 brainsci-13-01330-f002:**
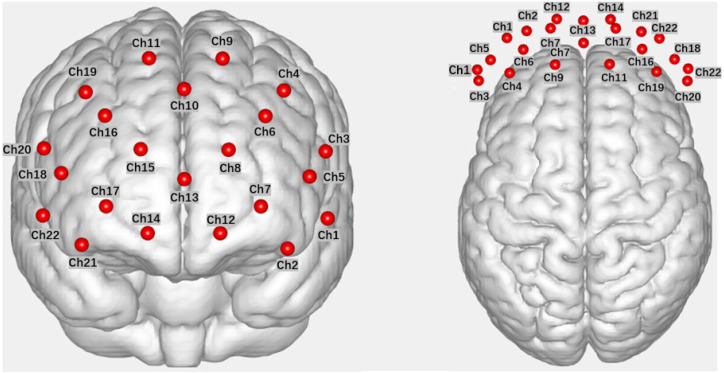
Channel layout diagram (front view, top view, in that order).

**Figure 3 brainsci-13-01330-f003:**
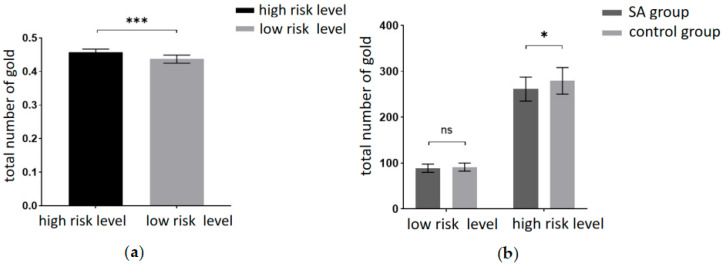
(**a**) Total number of gold coins at different levels of risk. (**b**) Total number of gold coins for the two groups of participants at different levels of risk (*** = *p* < 0.001, * = *p* ≤ 0.05, ns = no significant).

**Figure 4 brainsci-13-01330-f004:**
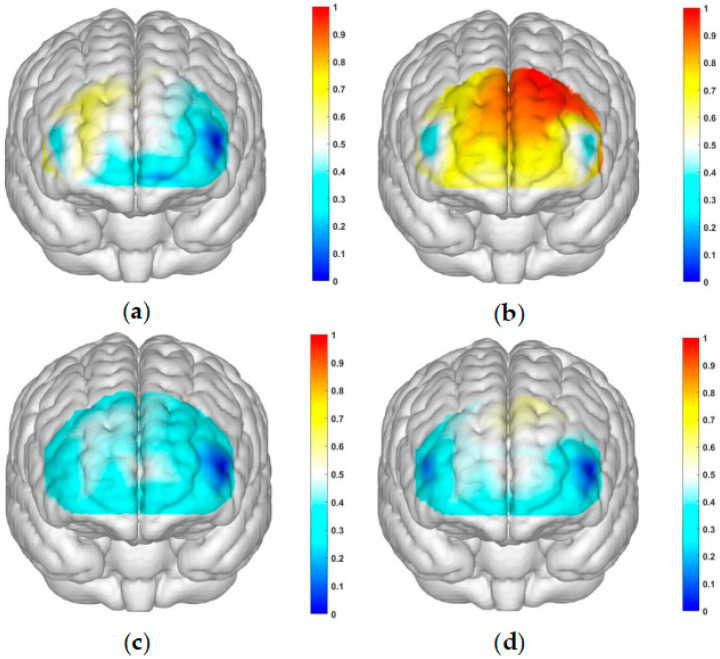
Activation maps of the measured brain regions, OFC and dlPFC, within the addiction group. (**a**) Represents the addiction group under condition 1: high-risk level with a gain outcome; (**b**) represents the addiction group under condition 2: high-risk level with a loss outcome; (**c**) represents the addiction group under condition 3: low-risk level with a gain outcome; and (**d**) represents the addiction group under condition 4: low-risk level with a loss outcome.

**Figure 5 brainsci-13-01330-f005:**
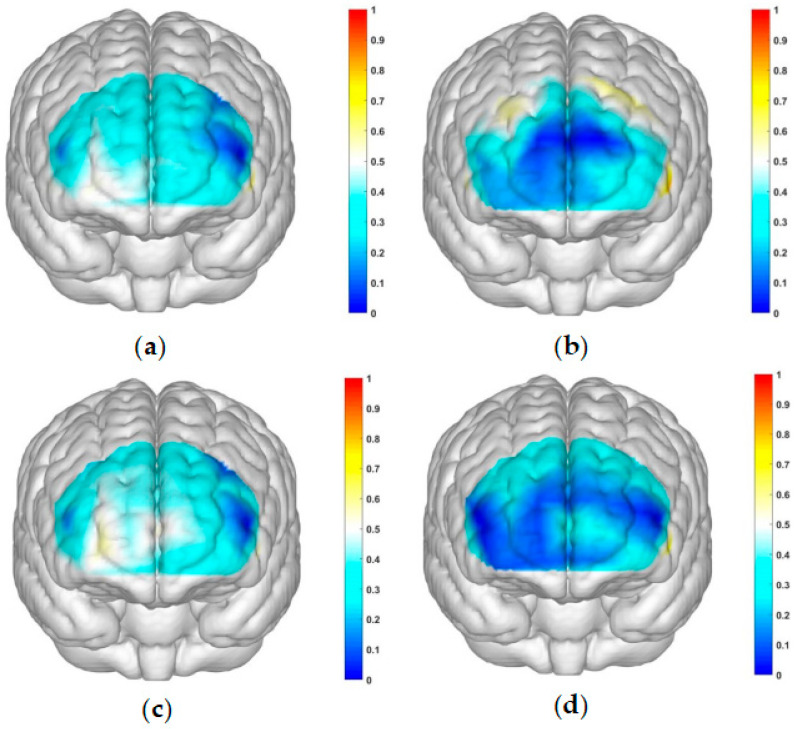
Activation maps of the measured brain regions, OFC and dlPFC, within the control group. (**a**) Represents the control group under condition 1: high-risk level with a gain outcome; (**b**) represents the control group under condition 2: high-risk level with a loss outcome; (**c**) represents the control group under condition 3: low-risk level with a gain outcome; and (**d**) represents the control group under condition 4: low-risk level with a loss outcome.

**Figure 6 brainsci-13-01330-f006:**
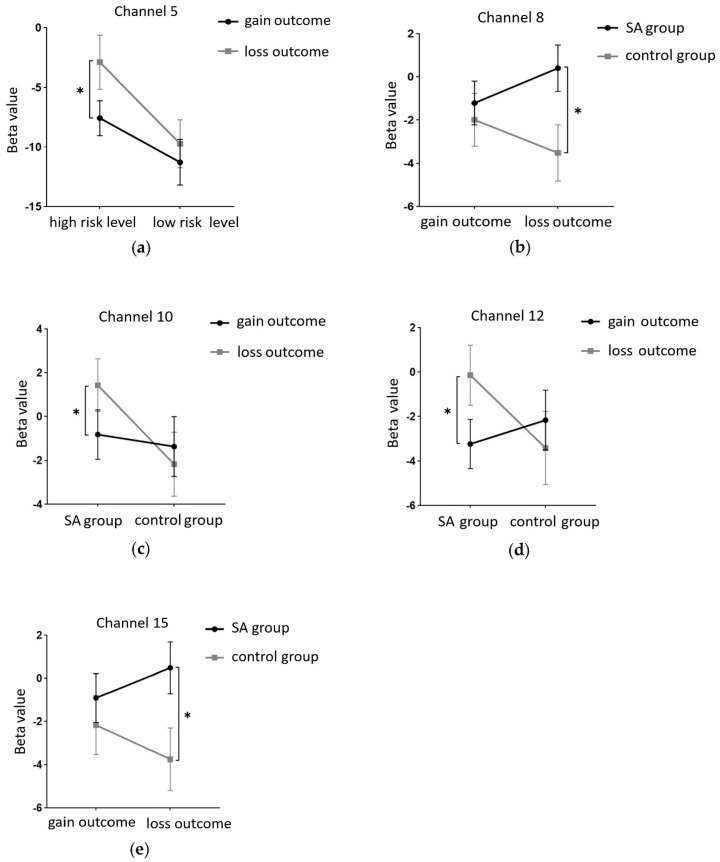
(**a**) Interaction between the amount of risk and the outcome in Channel 5; (**b**) Interaction between the group type and the outcome in Channel 8; (**c**) Interaction between the group type and the outcome in Channel 10; (**d**) Interaction between the group type and the outcome in Channel 12; (**e**) Interaction between the group type and the outcome in Channel 15 (* = *p* ≤ 0.05).

**Figure 7 brainsci-13-01330-f007:**
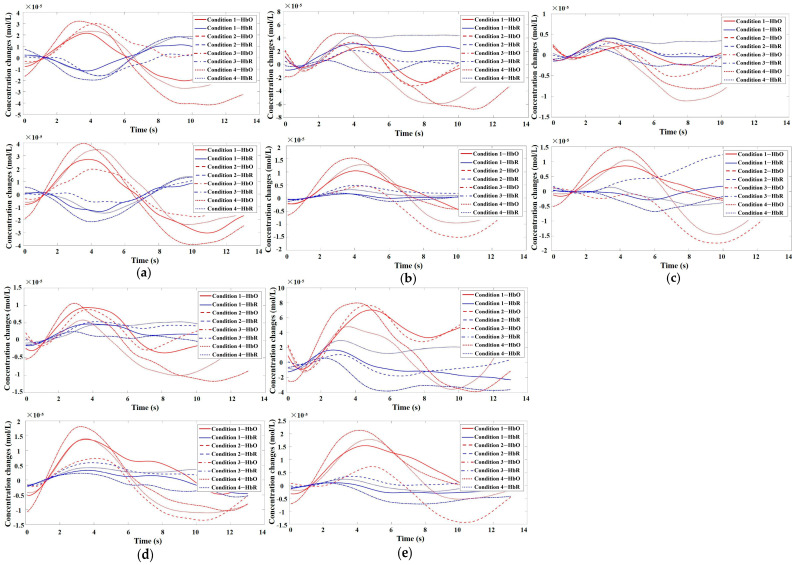
Time series plots of HbO/HbR for participants in Channels 5, 8, 10, 12, and 15. (**a**) Time series plot of HbO/HbR for participants in Channel 5. The top graph represents the addiction group, while the bottom graph represents the control group; (**b**) Time series plot of HbO/HbR for participants in Channel 8. The top graph represents the addiction group, while the bottom graph represents the control group; (**c**) Time series plot of HbO/HbR for participants in Channel 10. The top graph represents the addiction group, while the bottom graph represents the control group; (**d**) Time series plot of HbO/HbR for participants in Channel 12. The top graph represents the addiction group, while the bottom graph represents the control group; (**e**) Time series plot of HbO/HbR for participants in Channel 15. The top graph represents the addiction group, while the bottom graph represents the control group.

**Table 1 brainsci-13-01330-t001:** Demographic information and related scale scores.

	SA Group (M ± SD)	Control Group (M ± SD)	t	*p*
Age	19.92 ± 1.801	20.18 ± 2.976	−0.436	0.665
Gender	1.60 ± 0.500	1.76 ± 0.437	37.394	0.266
SAS-SV	45.20 ± 7.948	25.00 ± 3.317	34.475	≤0.001
SAS-C	77.2 ± 13.329	55.82 ± 8.911	5.781	≤0.001
Positive effects	25.88 ± 4.157	28.76 ± 5.154	−2.003	0.052
Negative effects	18.24 ± 5.166	14.12 ± 3.586	2.851	0.007
BIS	63.56 ± 9.247	59.06 ± 5.910	1.772	0.084
Attentional impulsivity	15.76 ± 2.728	13.29 ± 1.896	3.229	0.002
Movement impulsivity	22.20 ± 4.213	20.65 ± 3.372	1.267	0.212
Unplanned impulsivity	25.60 ± 3.958	25.12 ± 3.621	0.401	0.691
BDI	12.52 ± 10.215	2.53 ± 2.065	4.750	≤0.001
BAI	36.24 ± 10.199	27.24 ± 3.327	4.105	≤0.001

## Data Availability

The data presented in this review are available upon request from the corresponding author.

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
