# Peer review of "Exploring the Impact of Smartphone Addiction on Risk Decision-Making Behavior among College Students Based on fNIRS Technology"

_brainsci, 2023, doi:10.3390/brainsci13091330_

Round 1

Reviewer 1 Report

The manuscript by Liu et al addresses a highly important and contemporary topic, especially considering the widespread use of smartphones, with over 6.8 billion in circulation worldwide. The authors' discovery that individuals potentially addicted to smartphone use exhibit a significantly less efficient level of decision-making evaluation compared to non-addicted subjects is indeed intriguing. However, while this information is valuable, there are certain aspects throughout the manuscript that could enhance the overall quality of the work.

In the introduction, which provides a comprehensive overview of the topic, some attention to detail is needed. For instance, the abbreviation 'SA,' representing smartphone addiction, is redundantly defined multiple times, which could be streamlined. Additionally, on line 33, the reference to 'Kwon1' is somewhat unclear and could benefit from further explanation or context. Furthermore, the presentation of the two hypotheses (H1/H2) at the end of the introduction appears somewhat schematic. This style may be more suitable for a design-oriented manuscript and could be adjusted to provide a more comprehensive understanding of the research conducted.

In the methods section, the description of the groups to which the examined subjects belong is noteworthy, as is the schematic representation of the experiment. However, there appears to be a discrepancy regarding the first blank screen duration; it is mentioned as 60 seconds in the text but is not reflected in the schematic.

Regarding Figure 6, it would be beneficial to include not only the channels in the caption but also directly on the graphs themselves to enhance clarity. The current presentation can be somewhat confusing.

In the discussion section, delving into the significance of each channel, both anatomically and functionally, would add substantial value. However, the scientific discussion seems to lack enthusiasm. Instead of fostering a dialogue with existing literature to help readers grasp and reinforce the main message, it tends to repetitively state results without providing sufficient explanation or contextual value.

Considering the involvement of fundamental areas in the control of impulsivity in smartphone addiction (AS), the authors might reasonably speculate about its potential to lead to other forms of addiction or divergent social conditions. This could be an interesting avenue to explore further in the discussion.

N/A

Author Response

Thank you for taking the time to review our manuscript and for your valuable feedback. We have carefully considered each of your suggestions and have provided detailed explanations and justifications for them in our revised manuscript. Please find the specific modifications outlined below. Once again, we sincerely appreciate your input and suggestions.

  1. The abbreviation 'SA,' representing smartphone addiction, is redundantly defined multiple times, which could be streamlined.

Answer:thank you for your feedback, the issues have been addressed and corrected.

  1. On line 33, the reference to 'Kwon1' is somewhat unclear and could benefit from further explanation or context.

Answer:the reference in the first paragraph has been updated and the incorrect reference has been removed. Thank you for bringing this to our attention.

  1. The presentation of the two hypotheses (H1/H2) at the end of the introduction appears somewhat schematic. This style may be more suitable for a design-oriented manuscript and could be adjusted to provide a more comprehensive understanding of the research conducted.

Answer: thank you for your valuable feedback. Detailed revisions have been made in the manuscript to address the issues you raised.

  1. There appears to be a discrepancy regarding the first blank screen duration; it is mentioned as 60 seconds in the text but is not reflected in the schematic.

Answer: thank you for pointing out the issue. The feedback has been taken into consideration and the necessary revisions have been made in the manuscript.

  1. Regarding Figure 6, it would be beneficial to include not only the channels in the caption but also directly on the graphs themselves to enhance clarity. 

Answer: thank you for your suggestion. We have fully accepted it and made the necessary modifications to Figure 6.

  1. In the discussion section, delving into the significance of each channel, both anatomically and functionally, would add substantial value. However, the scientific discussion seems to lack enthusiasm. Instead of fostering a dialogue with existing literature to help readers grasp and reinforce the main message, it tends to repetitively state results without providing sufficient explanation or contextual value.

Answer: thank you for your valuable suggestion. We completely agree with it and have made targeted modifications in the manuscript based on your feedback.

  1. Considering the involvement of fundamental areas in the control of impulsivity in smartphone addiction (AS), the authors might reasonably speculate about its potential to lead to other forms of addiction or divergent social conditions. This could be an interesting avenue to explore further in the discussion.

Answer: thank you for your input. We fully agree with your opinion and have added this aspect to our manuscript for future research. 

Reviewer 2 Report

The authors of this manuscript made an interesting study mixing behavioral variables with brain imaging techniques. The article shows a good writing process.

I suggest to the authors the next:

1. In the Methods section (page 3, fourth paragraph), please explain and justify the type of sampling used.

2. On page 4, experimental tools, the authors should provide evidence about the validity and consistency of the scales used, in China´s context.

3. On page 4, last paragraphs, the authors stated that they used the Beck depression and anxiety scales; also the Barrat Impulsiveness Scale. Please be specific, explain, and justify the parametric properties of these scales in China´s context.

4. On page 5, first paragraph, the author must explain the way participants were divided and assigned to the groups.

5. On page 7 (results), first paragraph, I kindly suggest to the authors to delete the reporting of no significant differences in the text. That is not relevant to the study.

6. On page 7, Table 1, Please, change the 0.000 p-values, to ≤ 0.001

7. On page 11, Figure 7, please check if the scales used are right (different levels of scale for each plot are depicted).

Author Response

Thank you for taking the time to review our manuscript and for your valuable feedback. We have carefully considered each of your suggestions and have provided detailed explanations and justifications for them in our revised manuscript. Please find the specific modifications outlined below. Once again, we sincerely appreciate your input and suggestions.

  1. In the Methods section (page 3, fourth paragraph), please explain and justify the type of sampling used.

Answer: thank you for your valuable comment. We appreciate your input. In response to your suggestion, we have made improvements in the method section of the manuscript. Specifically, we have provided a clear explanation that participants were recruited through random sampling.

  1. On page 4, experimental tools, the authors should provide evidence about the validity and consistency of the scales used, in China´s context.

Answer: we appreciate your valuable comment. In response to your suggestion, we have added relevant literature to support the applicability of each scale in the experimental tools section, specifically in the context of China.

  1. On page 4, last paragraphs, the authors stated that they used the Beck depression and anxiety scales; also the Barrat Impulsiveness Scale. Please be specific, explain, and justify the parametric properties of these scales in China´s context.

Answer: thank you for your suggestion to specifically explain and justify the parametric properties of these scales in the context of China. In our manuscript, we have provided a brief explanation and supplemented it with references to three studies that are applicable to research conducted on Chinese populations.

  1. Sun, X. Y.; Li, Y. X.; Yu, C. Q.; Li, L. M. Reliability and validity of depression scales of Chinese version: a systematic review. Zhonghua Liu Xing Bing Xue Za Zhi 2017, 38(1), 110-116.
  2. Liang, Y. Depression and anxiety among elderly earthquake survivors in China. J Health Psychol 2017, 22(14), 1869-1879.
  3. Liang, Y.; Wang, L.; Zhu, J. Factor structure and psychometric properties of Chinese version of Beck Anxiety Inventory in Chinese doctors. J Health Psychol 2018, 23(5), 657-666.
  4. On page 5, first paragraph, the author must explain the way participants were divided and assigned to the groups.

Answer: Your question is very important to this study, and we have provided a detailed explanation in our manuscript.

  1. On page 7 (results), first paragraph, I kindly suggest to the authors to delete the reporting of no significant differences in the text.

Answer: thank you for your suggestion. We have carefully considered your advice and agree that it may be more appropriate to remove the reporting of no significant differences in the text. By doing so, we can streamline the presentation of our results and avoid potential redundancy. Therefore, we have made the necessary revisions and removed the reporting of non-significant differences from the first paragraph.

  1. On page 7, Table 1, Please, change the 0.000 p-values, to ≤ 0.001.

Answer: thank you for your suggestion. We have made the necessary modifications in the table as per your comment.

  1. On page 11, Figure 7, please check if the scales used are right (different levels of scale for each plot are depicted).

Answer: thank you for pointing out the issue. We have made the necessary adjustments to the scales in Figure 7. The purpose of this figure is to illustrate the changes in blood oxygenation in two groups of participants under different paradigm conditions, where response differences exist in specific channels. This figure does not include any statistical tests and is intended to provide readers with a visual representation of the changes in blood samples in a more intuitive manner.

Reviewer 3 Report

In this paper, the authors aim to investigate "whether smartphone addicts display different decision-making behaviors and brain neural activity at varying risk levels within a risky decision-making context".

I can identify novelty in this study, and I think it reads technically fine.

Suggestions and questions (answers can/should be used to improve the paper):

1. All abbreviations must be declared the first time in the text, including in the abstract (e.g., fNIRS, dlPFC)

2. line 33: Kwon1 characterizes Smartphone Addiction (SA) as.. -> Kwon et al. [1, 2] characterizes Smartphone Addiction (SA) as.. - in this case, reference 3 should not be cited.

3. Section 2.3.7. has no heading.

4. Main concern: It is not clear for me why the experimental task is a "risk decision-making task". Also, to understand the dependent variables, this task should be better described. For example, what is encountering a ghost? How to obtain a gold coin?

5. Avoid paragraphs with only one sentence.

6. Limitations of the study should be acknowledged.

7. Future work could be provided.

Author Response

Thank you for taking the time to review our manuscript and for your valuable feedback. We have carefully considered each of your suggestions and have provided detailed explanations and justifications for them in our revised manuscript. Please find the specific modifications outlined below. Once again, we sincerely appreciate your input and suggestions.

  1. All abbreviations must be declared the first time in the text, including in the abstract (e.g., fNIRS, dlPFC)

Answer: Thank you for your feedback. We apologize for the oversight in not declaring all abbreviations. we have made sure to declare all abbreviations, including those used in the abstract, in the revised version of our manuscript.

  1. Line 33: Kwon1 characterizes Smartphone Addiction (SA) as.. -> Kwon et al. [1, 2] characterizes Smartphone Addiction (SA) as.. - in this case, reference 3 should not be cited.

Answer: thank you for your detailed explanation of the issue. We have made the necessary revisions to the manuscript.

  1. Section 2.3.7. has no heading.

Answer: thank you for pointing out the issue. This section has been revised and its content has been integrated into other sections.

  1. Main concern: It is not clear why the experimental task is a "risk decision-making task". Also, to understand the dependent variables, this task should be better described. For example, what is encountering a ghost? How to obtain a gold coin?

Answer: thank you for raising the question about the experimental paradigm. The experimental task is called a "risk decision-making task" because participants are required to make decisions on whether to open the box in different risk scenarios. The risks are divided into different levels, and participants must assess the level of risk before deciding whether to open the box. As for the dependent variables, encountering a ghost refers to the appearance of a ghost in the game, which is a negative outcome. Obtaining a gold coin, on the other hand, is a positive outcome. These variables are relevant to the study's objective, which is to investigate the effect of risk perception on decision-making. Additionally, we have included supplementary explanations in the program section that provide further details on these variables.

  1. Avoid paragraphs with only one sentence.

Answer: thank you for your feedback. We have made the necessary revisions to the manuscript by merging the sentence paragraphs.

  1. Limitations of the study should be acknowledged.

Answer: thank you for your suggestion. We have made the necessary revisions to the manuscript by addressing the issues related to the limitations of the study.

  1. Future work could be provided.

Answer: thank you for your valuable feedback. We have added a section in the manuscript to discuss future research directions.